# "We can't do without it": Parent and call-handler experiences of video triage of children at a medical helpline

Caroline Gren [1,2]*, Ingrid Egerod[2,3], Gitte Linderoth[2,4,5], Asbjoern Boerch Hasselager[6], Marianne Sjølin Frederiksen[6], Fredrik Folke[2,4,7], Annette Kjær Ersbøll[4,8], Dina Cortes[1,2], Hejdi Gamst-Jensen[9,10]

1 Department of Pediatrics and Adolescence Medicine, Copenhagen University Hospital - Amager and Hvidovre, Copenhagen, Denmark, 2 Department of Clinical Medicine, University of Copenhagen, Copenhagen, Denmark, 3 Department of Intensive Care, Copenhagen University Hospital - Rigshospitalet, Copenhagen, Denmark, 4 Copenhagen University Hospital - Copenhagen Emergency Medical Services, Copenhagen, Denmark, 5 Department of Anesthesia and Intensive Care, Copenhagen University Hospital - Bispebjerg and Frederiksberg, Copenhagen, Denmark, 6 Department of Pediatrics and Adolescence Medicine, Copenhagen University Hospital - Herlev and Gentofte, Copenhagen, Denmark, 7 Department of Cardiology, Copenhagen University Hospital - Herlev and Gentofte, Copenhagen, Denmark, 8 National Institute of Public Health, University of Southern Denmark, Odense, Denmark, 9 Department of Clinical Research, Copenhagen University Hospital - Amager and Hvidovre, Copenhagen, Denmark, 10 Department of Emergency Medicine, Copenhagen University Hospital Amager Hvidovre, Copenhagen, Denmark

* ida.caroline.gren.02@regionh.dk

**Data Availability Statement:** The video triage studies were conducted as a quality improvement projects and were approved by the local hospital managements and the Emergency Medical

## Abstract

### Background

Pediatric out-of-hours calls are common, as parents worry and seek reassurance and shared responsibility. Nevertheless, most children assessed in this context are not seriously ill. Conventional telephone triage lacks visual cues and is further limited by third part communication in calls concerning children. We investigated implementation of video triage in two previous studies. The aim of the present study was to investigate 1) How video triage versus telephone triage in children was experienced by parents and call-handlers, and 2) call-handlers' evaluation of the video triage projects.

### Methods

We triangulated data from surveys and interviews in five sub-studies. Sub-study 1: Parents' experience of video triage reported in closed-ended questionnaire items using quantitative analysis; Sub-study 2: Parents' experience of video triage reported as questionnaire free-text using qualitative content analysis; Sub-study 3: Call-handlers' experience of video triage reported in closed-ended questionnaire items using quantitative analysis; and Sub-studies 4 and 5: Individual interviews of call-handlers' experience of 1) video triage using thematic analysis and 2) the video triage project using process evaluation.

### Results

Most parents' comments regarding video triage were positive (n = 164, 83%). Video triage was perceived as reassuring and reducing the likelihood of misunderstandings and

Services. The studies were therefore not registered with the regional and/or national research boards, and written consent were not obtained from the participants. The studies was furthermore assessed by the Research Ethics Committee in the Capital Region of Denmark, who deemed that approval was not indicated. As such, sharing of sensitive patient data is prohibited by Danish legislation. With referral to the Acceptable Data Access Restrictions of PLOS, we have prepared anonymized minimal data sets, which can be accessed by reasonable request to nanna.birch. andersen@regionh.dk at Copenhagen Emergency Services.

**Funding:** The video triage studies were funded by the Danish foundation TrygFonden (ID 124362; awarded to ABH; www.tryghed.dk), the Research Foundation at Amager Hvidovre Hospital (no ID; awarded to CG; https://www.hvidovrehospital.dk/forskning/Sider/default.aspx) and the Research Foundation of the Capital Region (A6207; awarded to DC; https://www.regionh.dk/english/research-and-innovation/Pages/default.aspx). The funders had no role in study design, data collection and analysis, decision to publish, or the preparation of the manuscript.

**Competing interests:** The authors have declared that no competing interests exist.

unnecessary hospital visits. Call-handlers experienced that video triage improved patient assessment and caller reassurance. Some call-handlers complained that the time allocated for study participation was inadequate and requested a more accessible video set-up. Both parents and call-handlers were significantly more satisfied and reassured after video triage than after telephone triage and suggested video triage as a permanent option.

## Conclusion

Video triage was appreciated by parents and call-handlers and was recommended as a permanent option. The call-handlers suggested that designated time for participation in the studies would have been desirable in this busy call-center. We recommend video triage as a contemporary solution in out-of-hours service.

## Introduction

The organization of out-of-hours (OOH) services around the world is diverse and presents challenges that may lead to unnecessary hospital visits, incorrect treatment allocation and lack of continuity [1, 2]. In many countries, OOH call-handlers serve as gatekeepers to meet the increased demand on OOH care and to relieve general practitioners' (GP) workload. Telephone triage is safe [3] and callers are generally satisfied [4, 5]. However, the task of call-center staff is not easy and might be associated with caregiver/gatekeeper conflicts, difficult decision-making due to lack of visual cues and communicating through a third party [6–8].

There are many pediatric calls to OOH services [9], but these children are rarely seriously ill. In a Belgian study of 51 GPs, only 1% of children with acute signs of infections had a serious infection (defined as pneumonia, sepsis, meningitis, osteomyelitis and pyelonephritis) [10]. This is in line with findings from 188 million pediatric emergency visits in the USA where only 0.5% concerned serious conditions [11]. Nonetheless, parental worry is common when children are ill. Parents contact OOH services when they feel they have lost control, need to share responsibility or find reassurance [12–15]. Previous experiences with the healthcare system impact parents' help-seeking behavior profoundly. Increased parental empowerment after healthcare interaction may lead to increased parent involvement and improved symptom management [16]. Conversely, felt or enacted criticism during contact with health professionals makes the parents avoid such situations and may lower their self-efficacy in future episodes of illness [17].

Consequently, a tool that could optimize triage and enhance the parents' reassurance is desirable, and this may potentially reduce OOH hospital visits. A natural step towards a more efficient triage process would be to introduce video streaming at call-centers, as Internet access and technological progress advance globally. In 2019–2020 we investigated video triage of sick children in two studies at the OOH Medical Helpline 1813 (MH1813) at the Copenhagen Emergency Medical Services, Denmark. The primary aim was to investigate if significantly more children in the studies could stay at home after video triage because of optimized triage and increased parental reassurance. A group of nurse call-handlers participated in the two studies of children with airway symptoms or fever, respectively. The children were six months to six years of age in the airway study and three months to six years in the fever study. The quantitative results regarding patient outcome in the airway study are reported in a separate paper (reference to airway paper, if accepted).

The current study is part of the aforementioned studies and is, as such, an important part of the project evaluation of the studies and may aid the set-up of future video triage tools at medical helplines. The aim of the study was to investigate 1) How video triage vs. telephone triage in children was experienced by parents and call-handlers, and 2) to describe call-handlers' evaluation of participation in the video triage project.

## Methods

### Study design and context

We triangulated methods and data sources (parents and call-handlers) to improve the quality of our study. We surveyed parents and call-handlers regarding their experience of video triage vs. telephone triage, and we interviewed call-handlers about their experience of the video triage set-up process. The study group consisted of investigators with different competencies, genders, experiences and professional backgrounds to enhance our discussions and ensure maximum variation. This study is reported in accordance with Standards for Reporting Qualitative Research [18], as the qualitative analyses in this study outweighs the quantitative components.

The context of sub-studies 1–5 is the "Video triage projects" that were conducted prior to the present sub-studies in the Capital Region of Denmark where MH1813 handles acute illness outside GP's opening hours. MH1813 must be contacted for preassessment by a call-handler before presenting to a face-to-face consultation. MH1813 receives approximately 1 million calls annually and is predominantly staffed by registered nurses, who handle approximately 80% of the calls. The remaining 20% are handled by physicians, either directly or in calls forwarded by nurses. The nurses have a broad variance of employment experiences and receive a six-week introduction consisting of theoretical and practical training. The nurses must use an electronic triage tool that guides them in their line of questioning and responses. The physicians receive a shorter introduction, mainly concentrating on practical training of the computer software, and are not required to use the electronic triage tool. The first (CG) and last author (HGJ) have worked along the staff at the MH1813 but have not undertaken a clinical role within the organization.

Children aged 0–5 years are a highly prevalent group at MH1813. In 2019, almost 18% of all calls (180,000 calls) concerned this age group [19]. The call-handler basically has two possible responses in pediatric calls: 1) the child stays at home (self-care and contact a GP the next workday if needed) or 2) the child is referred to hospital (pediatric urgent care clinic or pediatric emergency department). In a life-threatening situation the call is forwarded to the Emergency Medical Services' ambulance service.

All study call-handlers participating in the video triage projects were registered nurses. Approximately 20 nurses were initially selected by the management as being suitable participants. They were experienced call-handlers, comfortable using the computer systems and the triage tool and had technical aptitude. If a nurse left the group, another was invited to join, so that approximately 20 nurses were always active participants.

The video set-up was provided by GoodSAM (https://www.goodsamapp.org/instantOnScene), where a web-based video link to the caller was provided when the call-handler sent an activation link by text message to the caller's smartphone [20]. Access to the video camera was obtained after acceptance. No application was needed. After each study call, the call-handler answered a survey regarding the call and sent an electronic survey to the participating parent. These post-call procedures took two to four minutes.

Prior to implementation of the video triage projects, the MH1813 management was made aware that the study calls would take longer than standard calls due to related research procedures such as completing surveys. An agreement stated that the local floor managers could

temporarily interrupt patient inclusion if there was a high volume of incoming calls. The project group tried to engage more call-handlers to expedite the study, but this was infeasible due to recruitment difficulties at the time of the study.

## Sampling and data collection

We conducted five sub-studies as illustrated in Fig 1. Sub-study 1: Parents' experience of video triage expressed as closed-ended items in the parental questionnaire. Sub-study 2: Parents' experience of video triage expressed as free-text in the parental questionnaire; Sub-study 3:

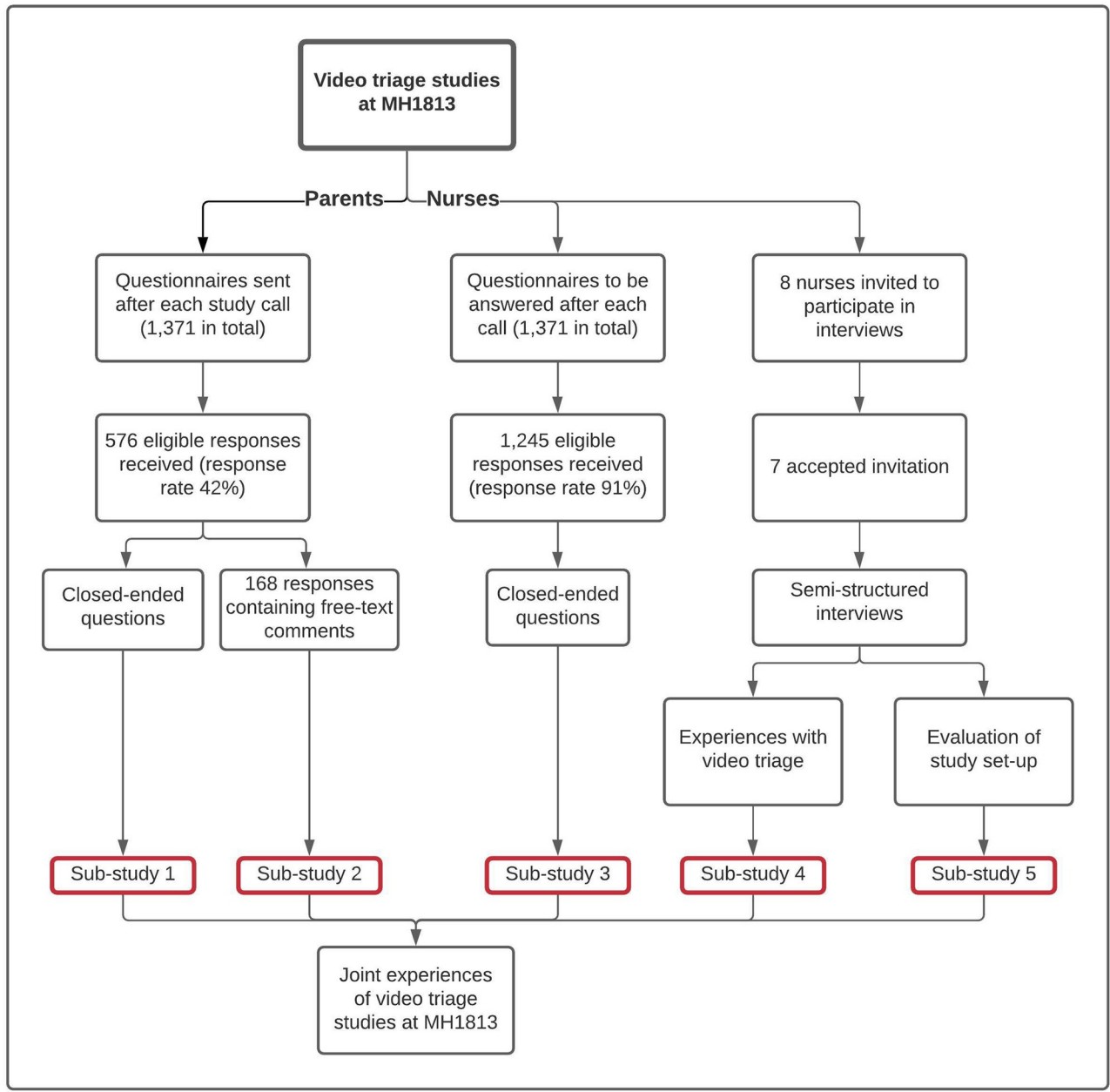

**Fig 1. Flowchart of data in the five sub-studies.** MH1813: Medical Helpline 1813.

Call-handlers' experience of video triage expressed as closed-ended items in the call-handler questionnaire, and Sub-studies 4 and 5: Call-handlers' experience of 1) video triage and 2) the video triage project described in individual semi-structured key-informant interviews.

**Sub-studies 1 and 2.** 1,371 questionnaires were sent via text message after each study call to the parents in the video triage group and the telephone triage group. A reminder was sent if no answer had been received within two days. The questionnaires investigated the parents' experiences of their call to MH1813, using eight closed-ended questions, mainly regarding satisfaction and reassurance (sub-study 1), S1 Appendix (S1A+B) items that were identical in the airway study and the fever study were included in the present study. Furthermore, the parents could leave comments in a text box and all free-text responses were included in sub-study 2. A total of 576 out of 1,371 responses (42%) were complete and eligible. Of these responses, 168 (29%) included a free-text comment, Fig 1.

**Sub-study 3.** The call-handlers were instructed to fill out a questionnaire after each call, and 1,245 (91%) questionnaires were returned complete and eligible for the study. We limited analysis to items that were identical in the airway and fever studies (S2A+B) in S2 Appendix.

**Sub-studies 4 and 5.** Eight call-handlers were invited and seven accepted participation in the interview study.

## Data analysis

Each sub-study was analyzed separately as different methods were applied. Data analysis for the different studies is elaborated below.

**Sub-studies 1 and 3 (quantitative studies).** The survey outcomes were described with frequencies (number, percentage). Differences between the video and telephone triage groups were analyzed using chi square-test, logistic regression or multinomial logistic regression when appropriate. The highest ranked survey response was compared with the other responses combined in the regression analyses, as there were few observations in several categories, which otherwise would yield unstable results with wide confidence intervals. An odds ratio (OR) was estimated for the logistic and multinomial logistics regression analyses with corresponding 95% confidence intervals (95% CI). A 95% CI for proportions was calculated by Wilson Binominal Proportion Confidence Interval. P-values less than 0.05 in two-sided tests were considered statistically significant. The statistical analyses were made with SAS Enterprise Guide 7.1 (SAS Institute, Cary, NC, USA), and Open Source Epidemiologic Statistics for Public Health (www.OpenEpi.com).

**Sub-study 2.** The parents' comments were analyzed using qualitative content analysis because we had little contextual knowledge of the parents and were unable to pose in-depth questions [21]. All free text comments were coded by meaning units, which were categorized and condensed in themes. As a meaning unit is defined as "words, sentences or paragraphs containing aspects related to each other through their content and context" [21], one comment could thus contain several meaning units. Finally, we compared the children's ages, participation in airway or fever study, triage responses and if the call was allocated to video triage or telephone triage in the eligible group of 168 calls with free text comment versus 1,203 non-eligible calls without comments. Finally, we compared the median age of the children with and without free text comments, and the number of free text comments in relation to symptom type, triage response, and type of triage (video or telephone).

**Sub-studies 4 and 5.** Individual semi-structured key-informant interviews were conducted during the last two months of the video triage studies. Participants were purposively selected to represent a varying degree of study activity. Sample size was estimated using the concept of information power [22], which is determined by a continuum within five

dimensions: aim (narrow vs. broad), specificity (dense vs. sparse), theory (applied vs. none), dialogue (strong vs. weak), and analysis (case vs. cross-case). The study had a narrow aim, experienced participants (dense specificity), knowledge of triage (applied theory), good investigator-participant communication (strong dialogue), and used cross-case analysis, all pointing toward a smaller sample size. We estimated that six to eight participants would be sufficient. As no considerable new information was added after the seventh interview, we assumed that information saturation had occurred.

The interviews were based on an interview guide developed by primary investigator CG, with subsequent revisions by other project members (DC, IE, HGJ), S3 Appendix. The experiences gained during the enrolment period of the main studies were used when formulating the interview guide. The interviews were conducted by CG who had in-depth knowledge of the overall project, as well as being well acquainted with the participants, which was considered an advantage in this case. The interviews were transcribed verbatim and analyzed using thematic analysis. Due to prior knowledge of the call-handlers and their work, we assumed it possible to be more interpretative and thus able to perform more data transformation in this study [23]. The statements regarding experiences of video triage were analyzed inductively and the responses regarding the overall project deductively.

**Sub-study 4.**   The process of analysis followed the steps described by Braun and Clarke [24]. After re-reading the transcripts several times, initial codes were created. Concerning the inductive analysis, the codes created were gathered in potential themes, which after multiple rounds of reviewing were adjusted, as well as checked for internal coherence within themes and clear distinction between themes. One code of great importance could be enough to form a theme. When writing the report, representative and exhaustive quotes were selected for each theme. The qualitative analyses were carried out using computer software NVivo 12 Plus (QSR International Pty Ltd., 2018). In addition, we calculated the median length and range of professional nursing experience and MH1813 employment of participants. The analyses were primarily performed by CG, in collaboration with DC, IE and HGJ, who all contributed in revising themes and interpretation of results.

**Sub-study 5.**   Our deductive theoretical framework for analysis was 'process evaluation' as described in 2002 by Steckler and Linnan [25]. This tool is constructed for public health interventions, presenting and defining key concepts of process evaluation to identify and explain which components of complex interventions work. Definition of included aspects is shown in S4 Appendix. The model describes: how the environment has affected the project execution *(context)*, the proportion of target audience participating *(reach)*, to what extent the participants have delivered the intervention *(dose delivered)*, the extent of participants' engagement with the intervention *(dose received)*, if the intervention was delivered as planned *(fidelity)*, the extent of implementation *(implementation)*, and procedures of attracting participants *(recruitment)*.

## Ethical considerations

The Research Ethics Committee in the Capital Region of Denmark deemed that approval was not indicated for the two video triage projects (Journal numbers: H-18049733 and H-19037554). The studies were registered at ClinicalTrials.gov (Id: NCT03874520 and NCT04074239). The interview study was registered and approved by the local data registration center (Permission: P-2020-29).

## Data sharing

As mentioned above, the projects were assessed by the Research Ethics Committee in the Capital Region of Denmark, who deemed that approval was not indicated. The projects were

therefore not registered with the regional and/or national research boards and written consent was not obtained from the participants. The projects were conducted as quality improvement projects and approved by the local hospital managements and the Emergency Medical Services.,. As such, Danish legislation prohibits sharing of sensitive patient data, but an anonymized minimal data set can be made available upon reasonable request to nanna.birch. andersen@regionh.dk at Copenhagen Emergency Medical Services.

## Results

### Sub-study 1. Parents' experience of video triage expressed in closed-ended questionnaire items

582 responses were received and 576 were completed and eligible for analysis; 260 from the airway study and 316 from the fever study, Table 1. The response rate was significantly higher in the video triage group; 47% versus 37% (p<0.001).

Significantly more parents in the video triage group answered that they to a very high extent: were satisfied with the call (OR 1.78, 95% CI 1.26–2.51) and felt safe regarding the assessment of their child (OR 1.42, 95% CI 1.02–1.98). 94% of the parents from the video triage group wanted video calls as a permanent option when calling MH1813.

### Sub-study 2. Parents' comments of video triage in a questionnaire

**Demographics.** The age of children involved in the 168 surveys with free text comments and the 1,203 surveys without, was similar with a median age of approximately 1.5 years. The number of free text comments was similar in the airway symptom and fever group (32% versus 26%). More free text comments were seen if children were triaged to home care rather than hospital (66% versus 56%) and if children were allocated to video triage as compared with telephone triage (70% versus 49%).

**Results of content analysis.** The parents' free-text responses regarding video triage roughly fell into two themes: 1) Advantages to video triage, and 2) Challenges of video triage. We coded 164 meaning units as positive regarding video triage (83%) and belonging to the first theme and 34 as somewhat negative. The remaining 57 comments addressed OOH in general. Themes, codes and quotes from the survey are presented in Table 2.

In summary, parents favoring video triage felt reassured, were less worried about misunderstandings, and experienced video assessment as efficient. Parents appreciated that hospital visits could sometimes be avoided after video triage. Some parents called MH1813 because they wanted face-to-face hospital assessment. In such situations the video triage was regarded as inadequate. Some technological issues needed to be addressed for optimal video triage. Many parents wished video triage to be a permanent option.

### Sub-study 3. Call-handlers' experiences of video triage expressed in closed-ended questionnaire items

Call-handlers returned a completed questionnaire after 91% (1,245/1,371) of the calls, Table 3. The scores show that call-handlers were more often satisfied with the call to a very high extent after video calls than after telephone triage, (OR 1.95, 95% CI 1.45–2.63). When adding 'to a high extent' to the analysis, in order to add more responses to the telephone triage group, the OR increased to 2.08 (95% CI 1.55–2.78). In 98% of video calls the call-handlers felt, to some extent or better, more reassured with the plan for the child than they thought they would have done in standard telephone triage, Table 3.

**Table 1. Parents' questionnaire responses.**

| | Video triage, n = 332 | Telephone triage, n = 244 | p-value | OR (95% CI) |
|---|---|---|---|---|
| **Response rate** | 47% (332/708) | 37% (244/664) | <**0.001**[*] | |
| **Are you satisfied in general with the contact to MH1813 today?[§]** | | | **0.001**[**] | |
| To a very high extent | 71% (236/332) | 58% (141/244) | | 1.80 (1.27–2.54) |
| To a high extent | 24% (79/332) | 37% (91/244) | | 1.0 (ref) |
| To some extent | 4% (14/332) | 5% (11/244) | | |
| To a small extent | 0.3% (1/332) | 0.4% (1/244) | | |
| Not at all | 0.6% (2/332) | - | | |
| **Did you get answers to your questions after the call to MH1813 today?[§]** | | | 0.09[**] | |
| To a very high extent | 64% (212/332) | 57% (139/244) | | 1.34 (0.95–1.87) |
| To a high extent | 29% (97/332) | 37% (91/244) | | 1.0 (ref) |
| To some extent | 5% (18/332) | 5% (12/244) | | |
| To a small extent | 0.9% (3/332) | 0.8% (2/244) | | |
| Not at all | 0.6% (2/332) | - | | |
| **Did you feel safe about the assessment of your child?[§]** | | | **0.04**[**] | |
| To a very high extent | 56% (187/332) | 48% (116/244) | | 1.42 (1.02–1.98) |
| To a high extent | 32% (107/332) | 42% (102/244) | | 1.0 (ref) |
| To some extent | 9% (30/332) | 10% (25/244) | | |
| To a small extent | 2%% (5/332) | 0.4% (1/244) | | |
| Not at all | 0.9% (3/332) | - | | |
| **Did you feel safe about the plan for your child?[§]** | | | 0.05[**] | |
| To a very high extent | 61% (201/332) | 52% (128/244) | | 1.39 (1.00–1.94) |
| To a high extent | 30% (99/332) | 40% (98/244) | | 1.0 (ref) |
| To some extent | 8% (27/332) | 6% (15/244) | | |
| To a small extent | 0,9% (3/332) | 1% (3/244) | | |
| Not at all | 0,6% (2/332) | - | | |
| **In your opinion, should video calls be a permanent option at MH1813?[§]** | | | <**0.001**[#] | |
| Yes | 94% (313/332) | 61% (148/244) | | 7.61 (2.77–20.8) |
| No | 2% (5/332) | 7% (18/244) | | 1.0 (ref) |
| I don't know | 4% (14/332) | 32% (78/244) | | 0.65 (0.21–2.03) |

MH1813: Medical Helpline 1813; OR: odds ratio;

[*]Chi square test;

[**]logistic regression;

[#]multinomial logistic regression;

[§] percentage (number/total number); ref: reference.

## Sub-study 4. Call-handlers' experiences of video triage expressed in semi-structured interviews

**Demographics.** The seven participants had a median of 25 years (range 11–37) of nursing experience and 4.5 years (range 1.5–6) of call-handling experience. Fig 2 shows the variable enrollment activity of the participants.

**Results of thematic analysis.** We identified three themes: *1) Video triage improves patient assessment*, *2) Video triage requires preparation*, and *3) Video triage is the way of the future*. The subthemes and corresponding quotes are presented in Table 4. The themes and their most defining aspects are unfolded below. The identities of the nurses are abbreviated N1-N7 in the following quotes.

**Table 2. Parents' experience of video triage.**

| Theme | Code | Meaning unit |
|---|---|---|
| **Advantages of video triage** | | |
| *Categories:* | | |
| Video calls reduce worry | Fear of misunderstanding | 'The video call and the possibility for the nurse to actually see him (patient) *made us feel really safe. As parents it is sometimes hard to describe the state and breathing precisely.'* (VF40) |
| | Better assessment | |
| | Reassurance after video call | |
| Video calls increase understanding | Good initiative | 'It would make sense to introduce video calls as standard procedure, because assessments are individual. For instance, I might see something else than a professional would.' (TF14) |
| | Good experience | |
| Video calls are convenient | Avoiding hospital commute | 'It is incredibly nice to have your young child assessed on video instead of having to go out in a cold car and go to the hospital and so on.' (VF49) |
| Video calls represent contemporary technology | State of the art treatment | 'Furthermore, it is part of the technological development—about time!' (TF14) |
| **Challenges to video triage** | | |
| *Categories:* | | |
| Video calls are not suitable for all situations | Video good with some symptoms | 'The video call was a really good possibility, but as you can't see everything on a video you should still be sent to a hospital when it is something serious (...)' (VF24) |
| | Must not replace hospital visit at all times | |
| | Still worried after video call | |
| Video calls can be challenging | Technical issues | 'I'm not sure the extra time spent using video gives a more accurate medical assessment. All the general questions before the video was started could probably have been answered during the video observation of my child (...)' (VA27) |
| | Could be used more efficiently | |

The respondents are coded with two letters and a number: V or T for *video* or *telephone*, F or A for *fever* or *airways study* and a continuous number. MH1813: Medical Helpline 1813.

**Table 3. Call-handlers' survey responses.**

| | Video triage, n = 661 | Telephone triage, n = 584 | p-value | OR (95% CI) |
|---|---|---|---|---|
| **Response rate** | 93% (661/707) | 88% (584/664) | **<0.001*** | |
| **Are you satisfied with the call in general?** [§] | | | **<0.001**** | |
| To a very high extent | 23% (153/661) | 13% (78/585) | | 1.95 (1.45–2.63)** |
| To a high extent | 63% (418/661) | 62% (362/585) | | 1.0 (ref) |
| To some extent | 12% (77/661) | 22% (128/585) | | |
| To a small extent | 1.4%% (9/661) | 2% (11/585) | | |
| Not at all | 0.6% (4/661) | 0.9% (5/585) | | |
| **Did you feel more confident about the choice of response when you could see and hear the child, than in standard telephone triage?** [§] | | | | |
| To a very high extent | 22% (145/661) | N/A | | |
| To a high extent | 63% (417/661) | N/A | | |
| To some extent | 13% (88/661) | N/A | | |
| To a small extent | 0.9% (6/661) | N/A | | |
| Not at all | 0.8% (5/66)1 | N/A | | |

OR: odds ratio;

*Chi square test;

**logistic regression;

[§] percentage (number/total number);

ref: reference; N/A: Non applicable.

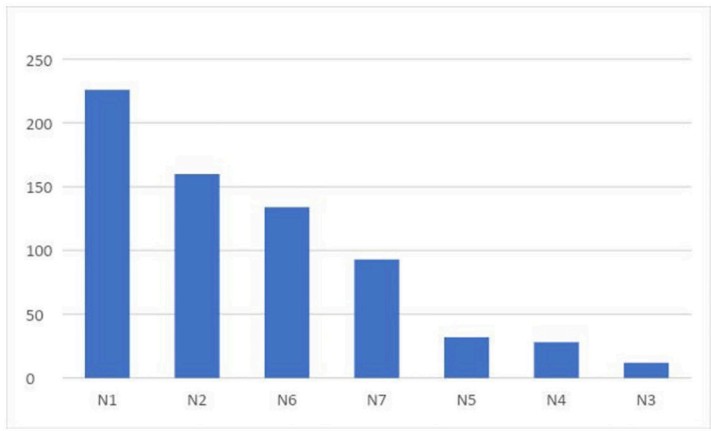

**Fig 2. Number of patients included by each call-handler.** N1-N7: Nurse 1 –Nurse 7.

*Theme 1*: *Video triage improves patient assessment*. According to the call-handlers, the parents and call-handlers used common terms differently. Call-handlers had to translate layman's terms to make them meaningful in a professional context, e.g. the terms 'drowsy' or 'difficulty breathing' had different meanings for parents and call-handlers. Interpretation became more challenging if there were cultural or language barriers as well. The call-handlers noticed that the degree of worry expressed by the parents often did not correspond to the severity of illness in the child. In these cases, it was difficult to reassure the parents and settle their worry. The call-handlers also noted that they identified several children that appeared more ill on video than the initial information presented by the parent had revealed.

> '*Well, when there's language barriers, you know, and you don't know how seriously ill these children are, right? That's where I really find that video calls clearly reveal how ill the children are, because the parents can't express it in other ways than through anxiety. (. . .) And the video calls are really, really good in those situations. Because then you see with your own eyes what it's about, right, and can uncover ABCD\* faster, than through poor communication, if you know what I mean?*'

*(N6)* (\*Algorithm used to systematically uncover acutely ill patients' condition)

Video calls bridge the gap between the perspectives of parents and call-handlers, because call-handlers can visually assess the child, rather than relying on parents' statements alone. Several call-handlers stated that the parents appeared more reassured after video calls. Some informants suggested that the call-handlers became more human when video was used, and they were not just a voice on the phone.

> '*And telling them, "well, I have seen this and that (. . .) and kind of saying 'now I have seen your child and I'm actually not worried". The parents feel much more reassured when they know that I have seen their child, than when just telling them what I have heard. Because they don't know how I work, they just think I'm some random person, but now I have seen and acknowledged their child, you know?*'

*(N5)*

**Table 4. Call-handlers' experiences of video triage.**

| Themes<br>*Sub-themes* | Codes | Quotes |
|---|---|---|
| **Video triage improves patient assessment** | Parents' worry | 'And I'm surprised that they (parents) *are so insecure about these normal things. Kids get sick, right*? *But it just gets so serious for them, somehow. It's just fever, you know*?' *(N3)* |
| | Parents' and call-handlers' different perspectives | |
| *Overcoming differences* | Communication | 'Yes, but what they say just doesn't fit the picture. I have been confirmed that we (call-handlers and parents) *speak different languages (. . .)*' *(N1)* |
| | Useful situations | '. . . *I remember that in the beginning, they said that they had poor contact with the child, and when I got it on video, the child was lying on his stomach watching iPad. It was like "wow", a wake-up call. And now I'm prepared, because I have seen it so many times.*' *(N1)* |
| *Commitment to care* | Reassurance for parents | 'But it has also been useful to reassure the parents, when I could see that it wasn't dangerous. So, I *reassure both the parents and me* (laughs*).*' *(N4)* |
| | Reassurance for call-handler | |
| | Parental guidance | '*I think that I, in collaboration with the parents, can explain what they should do over the next couple of hours and make a plan. Sometimes I, and I don't know if it's overkill, but after I've seen a child on video and made a plan, I'll call back in an hour to see if it worked.*' *(N4)* |
| *Building relationship* | Empathy | I think it's because they hear me communicate with them and describe their child and the color and breathing and having shown, what should I say, genuine interest in their child. They get more respect for my professional assessment. (. . .) They seem to have an easier time accepting it, because we have a different way of communicating, you know. The children are cute, and I can get emotional looking at them. (. . .) And that empathy also means something, and we communicate better. So yes, they seem happier.' *(N6)* |
| *More effective call-handling* | Higher or lower grade of urgency assessment | '*Int.: So, you think that occasionally, you're able to give the children a lower triage grade, so they'll stay at home or go to their GP the next day*? |
| | | *Nurse: Yes, I do that with almost every child with a fever.*' *(N1)* |
| | Useful situations | '*Nurse: But there was this one time when I didn't think the child was as distressed as it was as when I got to see it. And then I could really see* (mimicking heavy breathing*), you know, retractions all the way up to the neck, and the stomach going in and out.* |
| | | *Int.: And the parents couldn't express that*? |
| | | *Nurse: Not really, they said "but we're not doctors", they always say that* (laughs*), and maybe, maybe I hadn't asked properly, but I always ask "do they use their whole stomach to breathe?", but the child was actually a bit pale too.*' *(N4)* |
| | Duration of calls | '*I think it* (call duration*) remains the same, I think it sometimes gets shorter when I get them on video, because then (. . .) I write "video seen", and I can see that there is no rash, no neck stiffness, good contact, normal breathing.*' *(N1)* |
| | Gatekeeping | '*And that could be one of the advantages of this* (video triage*), that you're able to keep them* (children) *home. It's not a good idea that as many children as possible go to the pediatric emergency department, and maybe get infected by others. You can probably, and I have done that, eliminate some of them.*' *(N6)* |
| **Video triage requires preparation** | Difficulties of video triage | '*But it's hard when the children are sleeping. When they are tired already. So, when you get past a certain time of night, it almost doesn't make sense with video calls, in my experience. . .*' *(N2)* |
| *Unprepared callers* | Getting an extra sense | '*Because now I had to both watch and listen, whereas I'm used to just listening, right?*' *(N5)* |
| *New way to work* | Video streaming tool | '*Yes, sometimes it is, as I said, bad image quality, and sometimes the parents stand like this* (moving arms from side to side) *so I'll have to tell them to hold still.*' *(N1)* |
| *Technical issues* | | |
| **Video triage is the way of the future** | Second-hand information | '*It's always like that when someone is interpreting something, you always miss out on something. And it is always affected by what the other person thinks and how worried they are, because if they have experienced something before, or they have read about a disease or something else (. . .) So when it's another person calling, that could be good, yeah.*' *(N2)* |
| *Other useful situations* | Interpretation of symptoms | '*I think it would be the same, just with older people. Things like contact and breathing are important, because many things are just facts. "Pain, yes or no?" I don't need to see that on video, right? (. . .) if you suspect a stroke, it could be like "can you make a fist with both hands?", but I think it mostly would be breathing and contact.*' *(N7)* |
| *Video should be permanent* | Future video streaming possibility | '*Yes, for sure. I hope it's the future (. . .)* **We can't do without it**. *It will probably be a bigger part of our work. It just has to be easy to use, because we're answering so many calls. What's important is to avoid misunderstandings and such.*' *(N6)* |

*(Continued)*

**Table 4.** (Continued)

| Themes | Codes | Quotes |
|---|---|---|
| *Sub-themes* | | |
| *Risk of unnecessary use/ demand* | Future video streaming possibility | *'But it can give the opposite as well, I think. Because a lot of people immediately say, "can't we send a picture?". They don't start by telling me what it's all about. (…) And I think that can happen with a lot of parents, being as they are these days. (…) That could be a drawback. It could get time consuming* (handling unnecessary video demand).*' (N4)* |

N1-7: Nurse 1–7; Int.: Interviewer; MH1813: Medical Helpline 1813.

*Theme 2*: *Video triage requires preparation*. Video triage added another dimension to patient assessment but required some preparation from both parents and call-handlers.

'*It's an unfamiliar tool, because you are used to using your senses in a certain way and you are kind of visualizing what you are told.*'

*(N3)*

The parent should prepare by undressing and positioning the child for the call. The call-handlers should prepare by familiarization with the technical aspects of video triage, which requires frequent video activity. According to the call-handlers with lower inclusion rates, the technical set-up was complicated and sometimes provided inadequate image quality.

*Theme 3*: *Video triage is the way of the future*. According to all call-handlers, video calls should be a permanent option at MH1813. It was emphasized that it must be easily accessible and fast to use. It was suggested that the caller could be prepared during the introductory information before being connected to the call-handler. The call-handlers found video calls useful in patients with airway symptoms and fever and suggested other beneficial situations, such as for all third-party calls, when determining severity in trauma patients and when assessing complex symptoms where some form of interpretation is needed. Calls from individuals with non-Danish background would potentially benefit from video triage, due to language and culture barriers. Video streaming could help the call-handler to reach a more objective understanding of the sick person.

'*Yes, for sure, language problems would be good, and also because consultations with doctors are conducted differently in different parts of the world. And you describe your symptoms differently. And in some countries, you are used to exaggerate to be sure to get assessed, and in those situations it* (video streaming) *would be good for triage. What is really going on, right?*'

*(N5)*

In conclusion, the idea and experience of video triage were positive among the parents and call-handlers, with both emphasizing a higher degree of reassurance, and call-handlers stressing the possibility of optimized communication and triage.

## Sub-study 5. Project evaluation

Codes and comprehensive quotes resulting from the deductive analysis based on Linnan and Steckler's process evaluation [25] are presented in Table 5, and the results of the analysis are elaborated below.

**Table 5. Call-handlers' evaluation of video triage studies.** Deductive findings according to Steckler and Linnan's process evaluation model [25].

| Component of process evaluation | Quote |
|---|---|
| **Context** | '. . . but that's the way it is when is busy around here, then we don't have the time to deal with it (video calls), but I wish that there was some space created, so we were allowed to do it, right? You get a bad conscious when you're doing it and you see red and yellow numbers (indicating long waiting time for callers), right? But they (management) also look a lot at how long people are waiting. I'm just thinking that it would have been nice to not be so stressed about it. Like, if you were selected for this project, you were protected.' (N6) |
| | 'Nurse: That we had more time and space, right? We are under pressure. We are still short on colleagues. We are under pressure. So, it is the time; time is the big perspective in here. Because we are monitored, and we must get our calls. Even when they say, "I know you're part of that (video triage study)", then you still get asked "Why is your cadence getting lower?", right? |
| | Int.: So, there's a mismatch between what we wished and the reality? |
| | Nurse: Yes.' (N5) |
| **Reach** | "I think it's discouraging. And I have heard several say "Well, I'm taking a break from that" and I just think "we shouldn't do that, we have to use it or else it's just me using it, and then we can't get it done". So, I think that's discouraging. I don't know if you could do anything different. I suppose you just have to inform about the importance of doing it correctly. I don't know.' (N1) |
| | 'Int.: So, the technical set-up, it has been a barrier to including patients? |
| | Nurse: Yes. |
| | Int.: Is it the biggest factor for you not having included that many patients? |
| | Nurse: Yes, I think you could say that. (N3) |
| **Dose delivered** | 'Int.: When you are at work and get a call that fits the inclusion criteria, do you always include it to the project? |
| | Nurse: Yes, I do. There have been a few, where they were really ill, where I like, ethically, didn't feel that I could include them. Where the parents were so frightened that I felt that now wasn't the time to ask them if they wanted to participate in a project. But besides than that, I have included everyone. |
| | Int.: Then there haven't been any other reasons, too busy, or what do I know? |
| | Nurse: No, I have included all that I could. Maybe there have been some that I forgot, but not on purpose.' (N2) |
| | 'Int.: When you are at work and get a call that fits the inclusion criteria, do you always include it to the project? |
| | Nurse: No, I haven't (laughs). |
| | Int.: And why not? |
| | Nurse: No, I haven't, either because of lack of time or because I have been supervising colleagues in training, so that's when I have opted out.' (N5) |

(*Continued*)

**Table 5.** (Continued)

| Component of process evaluation | Quote |
|---|---|
| Dose received | 'Nurse: Before we started, I didn't think about it at all, but once you get used to this tool, **you can't do without it**. |
| | Int.: No. |
| | Nurse: In the beginning, the parents just said 'retractions', and then we'd refer them to admission, but we don't need that now. So, there's a huge difference. In the beginning, I would also send ambulances because of retractions, so it's quite different. And when they say impaired contact, and then you see the child running around the house, on video, right? So, it's like another world. I can't do without it.' (N1) |
| | 'Int.: Could you tell me what you overall think about video triage? |
| | Nurse: I think it's good. |
| | Int.: Yes. Tell me why. |
| | Nurse: Yes, but it's because I very often experience that they tell me that their children are drowsy, and when you ask them, "what is drowsy?", they can't really explain it. "She's just down." (. . .) And then you see the child on video, just being tired because of the fever, but attentive and looking around and so on. I think that's the best part. That you kind of get an idea about the situation, right, because if they think that their child is supposed to be active all the time despite the fever, of course they'll think they're drowsy, right. (. . .) So, I feel like you're able to get your own view of things.' (N2) |
| Fidelity | 'Nurse: Sometimes I had to make two video calls in a row, because **I couldn't do without the video**, but then I took two telephone calls, so it evened out. |
| | Int.: Okay. So once in a while you made video calls even though it should have been telephone calls? |
| | Nurse: Yes, I had to do it a couple of times. |
| | Int.: Yes. Tell me why. |
| | Nurse: Because they said retractions. Or impaired contact. But I heard that the contact was okay, and then I needed to see it on video. Or they say difficulty breathing, and I can't hear it. So, it's mostly when they say there's retractions and I don't think there is. Or they say that the breathing is fine, but I thought there was something else. Anyway, **I haven't been able to do without it** a couple of times.' (N1) |
| | 'Int.: When you've been at work after being introduced to the project, and get a call that fits the inclusion criteria, have you always included it to the project? And if not, then why not? |
| | Nurse: I did in the beginning. (. . .) but it faded out for me. |
| | Int.: Tell me why. |
| | Nurse: Yes, well I think it was all that trouble with logging in to that GoodSAM (video set-up), and all that 'who are you' and 'press this', and those pictures you had to, you know, tick (laughs), I found that really annoying, so I just didn't bother.' (N3) |

N1-7: Nurse 1–7; Int.: Interviewer.

**Context.** Enrolment was prolonged due to an insufficient number of call-handlers. Call-handlers were stressed by trying to maintain their regular call quota during the study, although this was not a united experience. They felt both internal stress and thus refrained themselves from including patients in high-flow periods, and they also experienced being asked by their leaders why their cadence went down. Dedicated study call-handlers were suggested in future studies. Finally, the call-handlers experienced inadequate management support, and wanted extra time allocated to participate in the research projects.

**Reach.** The reach was lower than predicted due to a lower inclusion rate than expected. The environment was experienced as stressful and the technical set-up sub-optimal by some participants, whereas others did not experience any technical difficulties or difficulties reaching their call quota.

**Dose.** The *dose delivered* was high for call-handlers that felt familiar with the video tool and who experienced little or no negative impact on their overall work performance. These nurses had a high inclusion rate. Some call-handlers failed to comply with the allocation strategy, as they switched to video if the symptoms appeared serious or the communication was challenging—these calls should have been excluded due to protocol violation. The *dose received* appeared high, because the call-handlers were all in favor of video triage.

**Fidelity.** Fidelity was impaired by inconsistent delivery and failing to include all eligible calls.

**Implementation.** Implementation varied among call-handlers due to the abovementioned issues of reach, dose delivered and fidelity.

**Recruitment.** Initial recruitment was done by managers who handpicked call-handlers that were assumed to have technical acumen. Performance, however, varied among the participating call-handlers.

In conclusion, all call-handlers were all positive towards video triage, but the local setting was not optimally prepared for the study. The primary challenge was that several call-handlers felt unable to participate in the study as well as reaching their expected patient quota. This put pressure on the call-handlers that tried to satisfy both roles.

## Discussion

Our main finding was that video triage of young children in a medical call-center was well received by parents as well as call-handlers and that both groups found it reassuring and wanted it as a permanent option. To the best of our knowledge, there have been no reports of the user's perceptions of video triage of sick children at a medical helpline. However, other areas of pediatric research have studied video-based communication. Video-based triage at low risk birth centers in Japan increased the sensitivity and specificity for predicting newborns with critical respiratory dysfunction prior to transfer to a specialized center [26]. A Danish study on neonatal tele-homecare showed that telehealth service provided the parents with increased reassurance and empowerment [27]. Video transmission of children with asthma exacerbation from parents' smartphones in a hospital waiting area to health care staff was feasible and appreciated by parents and health professionals [28]. The American Academy of Pediatrics has encouraged increased use of telemedicine to enhance access to pediatric care and address physician shortages [29]. Seen in the light of the ongoing COVID-19 pandemic, telemedicine has obvious advantages in restricting physical contacts, and increased use of telemedicine tools has indeed been encouraged [30, 31].

### Parents' experiences of video triage

Parents were reassured by video triage because their concerns and observations were shared directly with a healthcare professional. This alleviated misunderstandings and misinterpretation of information. These findings are confirmed by the quantitative analyses of the questionnaires, where parents expressed a higher degree of satisfaction, and importantly, of reassurance regarding assessment and plan. Moreover, parents experienced that video triage sometimes could save them a visit to the hospital, which was perceived as an advantage. Our findings are not surprising as prevention, diagnosis and reassurance are the primary expectations of parents contacting primary care due to acute illness, while treatment and cure are secondary [14]. Parents want clear information and counseling on how to care for their sick child and wish to share the responsibility with professionals [14, 32, 33]. This helps them feel reassured and better able to care for their sick child during the current and future episodes of illness [13, 15, 32]. Consequently, if the contact to a medical helpline relieves and empowers

parents by means of video streaming, the number of hospital visits might be reduced. Finally, as parents of young children themselves have grown up with digital communication, it is likely that the use of video streaming is familiar and reassuring even in the potentially stressful situation of caring for a sick child. The increasing use of modern telecommunication in societal services such as medical helplines does not seem disturbing for these parents; rather it appears welcomed.

## Call-handlers' experiences of video triage

The call-handlers agreed that parents and professionals speak different languages; figuratively speaking, and that video calls could decrease the risks of misunderstandings and erroneous triage; both under- and over-triage. Strikingly, 98% stated to feel more reassured about triage outcome after video triage than as expected if it had been standard telephone triage. Video calls were recommended for all third-party calls, in children as well as adults, i.e. in calls where information is conveyed by someone else than the patients themselves. Moreover, call-handlers believed that video calls were beneficial for parents with non-Danish background because of language barriers or cultural differences in perceptions of symptoms. The latter perception might be a consequence of ethnocentric views among nurses of Danish origin which may lead to inappropriate patient care, if they are not sufficiently aware and respectful of intercultural discrepancies in help-seeking behavior [34]. If video calls could help overcome some of the potential challenges of intercultural communication, it would be of importance, possibly alongside an increased focus on differences in help-seeking behavior.

Furthermore, call-handlers found it easier to reassure the parents when they used video streaming. A higher grade of perceived empathy was suggested, as this form of communication appears more personal. Indeed, in a framework of empathy in computer-mediated interactions, video calls seem to have a higher chance of evoking empathy in the care provider, since video streaming has higher degrees of verbal and non-verbal information than other media types, such as telephone calls [35]. When coupled with the immediacy of communication in video calls, the socioemotional cues of the caller can efficiently be transmitted to the care provider, who then can convey their empathy back to the caller. Moreover, empathy plays an important role in forming therapeutic alliances and building relationships [35, 36]. The factors discussed above relating to the parent-provider relationship, alongside parental feelings of being heard and seen have been identified as antecedents of parental empowerment. It may lead to positive consequences such as better involvement in care decisions, symptom management and advocacy for the child [16]. Both call-handlers and parents were found to be significantly more satisfied with the call after video triage, and this might reflect that the aspects discussed above such as feeling seen and heard and that the therapeutic relationship improved when using video as part of the contact.

## Call-handlers' evaluation of video triage studies

The project set-up influenced the experience of video triage and had some challenges. Varying activity levels and failure to adhere to the allocation strategy were problems discussed in the interviews, where lower inclusion rates were mainly caused by the stressful work environment and technical set-up. However, clinical research is inherently affected by a stressful environment and time constraints unless substantial resources are allocated to the study. The research group was aware of some of these challenges beforehand and tried to alleviate them, but since the studies were carried out in a functioning, busy call-center, we had to accept issues regarding bustle and stress. Nevertheless, the research group preferred to test this potentially useful tool under these real, clinical circumstances, rather than not at all, and the outcomes do indeed

reflect the reality of a busy call-center. Some call-handlers would have preferred to have been excused from their ordinary duties and to experience stronger management support. Other nurses found the video equipment easy to use and were not affected in their daily obligations. Despite these differing experiences, all nurses were positive to the idea of video triage and all wanted it as a permanent option, albeit in a form that was easier to access, and faster to use.

The call-handlers' perception of the intervention (video triage) possibly had a direct effect on the quantitative outcome measures of the project, which will be reported separately (reference to airway study, if accepted). There was some skewness in the inclusion of calls, where the call-handler could not do without video triage in some instances, and therefore bypassed the inclusion algorithm, which inherently contributed to the quantitative outcomes. Despite continuous monitoring of the call-handlers' fidelity to the set-up, we acknowledge that the context of a busy medical helpline was a challenging setting, as call-handlers were inclined to use the triage mode they personally favored over an inclusion algorithm. The project was in a sense smothered by its own success as the video streaming was easily adopted by most participants who then used it in an inappropriate way, which may influence the findings that should prove the success.

Although the experiences of video triage were positive and the study itself was well received by both management and participants, it proved difficult to implement research studies in the reality of a busy call-center. Ideally, better staffing and a better computer set-up should have been in place before the study was initiated. This would have ensured optimal conditions for success.

## Transferability and limitations

Call-centers vary regarding staff education and experience. In this study, the call-handlers were all nurses that had both clinical and call-handling experience, which may have influenced their views on video streaming. We lack information on what circumstances influenced the parents' views on video triage. The parents' overall questionnaire response rate was 42%, with a higher proportion of respondents having received video triage. Free-text comments were present in 29% of the responses, and these respondents had a higher frequency of allocation to video triage and referral to self-care at home. Consequently, these parents were expected to benefit the most from video triage, which might affect the generalizability of the findings. We did not seek to generalize in the content analysis of free-text comments. However, parents calling regarding sick children is a highly prevalent group of callers in all OOH settings [9], and since few children are seriously ill [10, 11] we assume that the present positive results towards video triage are applicable in most OOH settings regarding acutely sick children. Moreover, as many studies have reported that parents worry when their children are sick, a method that increases their reassurance is of great value. The call-handlers' experiences benefitted from a combination analytical and statistical generalization of results, using both closed-end questionnaire items and interviews.

Another limitation was that feasibility and fidelity towards the intervention were not sufficiently addressed prior to the trial. Even though the two video studies were quality improvement studies, they had nonetheless the characteristics of a complex intervention and as such certain steps could have been followed during the stages of developing, piloting and feasibility, evaluation and implementation [37]. If the two video studies had been executed as randomized clinical trials, it would have been a prerequisite.

## Conclusion

Video triage of children with fever or airway symptoms was valued by parents and call-handlers alike, who found the method reassuring and recommended it as a permanent option.

During the video triage study, the call-handlers needed more time allocated for research participation and requested a less complicated video set-up. We recommend video triage as a contemporary solution in out-of-hours service.

## Supporting information

**S1 Appendix. Appendix 1A+B.** Parents' questionnaires from the study regarding airway symptoms (Appendix 1A) and fever (Appendix 1B). CPR: Central Person Register (personal identification number).
(PDF)

**S2 Appendix. Appendix 2A+B.** Call-handlers' questionnaires from the study regarding airway symptoms (Appendix 2A) and fever (Appendix 2B). CPR: Central Person Register (personal identification number).
(PDF)

**S3 Appendix. Interview guide, used in semi-structured interviews with call-handlers.** MH1813: Medical Helpline 1813.
(PDF)

**S4 Appendix. Key process evaluation components (Steckler and Linnan, 2004).**
(PDF)

## Acknowledgments

A heartfelt thanks to all nurses participating in the video triage studies, they could not have been accomplished without you. The authors are especially grateful to the nurses participating in the interviews included in this study. To Head of Emergency Medical Dispatch Center; Marie Baastrup and CEO; Freddy K. Lippert in the EMS Copenhagen management for letting us perform this study at MH1813. To GoodSAM, for the usage of the Instant-on-Scene™ platform.

## Author Contributions

**Conceptualization:** Caroline Gren, Ingrid Egerod, Dina Cortes, Hejdi Gamst-Jensen.

**Formal analysis:** Caroline Gren, Ingrid Egerod, Annette Kjær Ersbøll, Dina Cortes, Hejdi Gamst-Jensen.

**Funding acquisition:** Caroline Gren.

**Investigation:** Caroline Gren.

**Methodology:** Caroline Gren, Ingrid Egerod, Dina Cortes, Hejdi Gamst-Jensen.

**Project administration:** Caroline Gren.

**Resources:** Caroline Gren.

**Supervision:** Ingrid Egerod, Annette Kjær Ersbøll, Dina Cortes, Hejdi Gamst-Jensen.

**Visualization:** Hejdi Gamst-Jensen.

**Writing – original draft:** Caroline Gren.

**Writing – review & editing:** Ingrid Egerod, Gitte Linderoth, Asbjoern Boerch Hasselager, Marianne Sjølin Frederiksen, Fredrik Folke, Annette Kjær Ersbøll, Dina Cortes, Hejdi Gamst-Jensen.

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
