## [Editor Report · Decision Letter 0]

26 Apr 2021

PONE-D-21-08760

“We can’t do without it”: parent and call-handler perceptions on video triage of children at a medical helpline

PLOS ONE

Dear Dr. Gren,

Thank you for submitting your manuscript to PLOS ONE. After careful consideration, we feel that it has merit but does not fully meet PLOS ONE’s publication criteria as it currently stands. Therefore, we invite you to submit a revised version of the manuscript that addresses the points raised during the review process.

This study reporting parental and call-handler perceptions about video triage would be significantly enhanced by including the quantitative data presented in the companion paper and presenting the investigation as a mixed-methods study. This would greatly support the generalizeability of the data.

We look forward to receiving your revised manuscript.

Kind regards,

Richard Bruce Mink

Academic Editor

PLOS ONE
---

## [Author Response · Author response to Decision Letter 0]

30 Jun 2021

Dear Richard Bruce Mink

On behalf of all the authors, I thank you for your interest in our manuscript “We can’t do without it”: parent and call-handler experiences of video triage of children at a medical helpline, PONE-D-21-08760.

We are today submitting a revised version, changed according to your points raised. Consequently, this manuscript now contains all data concerning the users’ experiences of video triage studies conducted at the Medical Helpline 1813, whereas the accompanying paper (Video triage of children with respiratory symptoms at a medical helpline is safe and feasible – a prospective quality improvement study, PONE-D-21-06235) solely focuses on patient outcome.

We have also revised the manuscript’s set-up, so it conforms with the style requirements. Furthermore, ast the sharing of sensitive patient data originating from quality improvement projects is not permitted by Danish law, anonymized minimal data sets have been prepared and are accessible on reasonable requests to the contact person mentioned in the manuscript.

This manuscript now conveys quite a rich picture of how the contemporary tool video streaming was received and experienced by the users, when being implemented at a functioning, busy call-center. Furthermore, the sub-study on project evaluation may aid other call-centers who are considering the implementation of a similar tool. We sincerely hope that the papers are accepted, as we believe that that may contribute with new and important results regarding health services, relevant to many health care providers globally.

Kind regards,

Dr. Caroline Gren

---

## [Decision Letter · Decision Letter 1]

14 Mar 2022

“We can’t do without it”: parent and call-handler experiences of video triage of children at a medical helpline

PONE-D-21-08760R1

Dear Dr. Gren,

We’re pleased to inform you that your manuscript has been judged scientifically suitable for publication and will be formally accepted for publication once it meets all outstanding technical requirements.

Kind regards,

Dylan A Mordaunt, MB ChB, MPH, MHLM, FRACP, FAIDH

Academic Editor

PLOS ONE

Additional Editor Comments (optional):

This now meets the criteria for publication. The comment made by the reviewer with regards the odds ratio, should be taken into account but I'll leave this to the authors to decide whether they modify their language and this isn't a prerequisite to publication.

Reviewers' comments:

Reviewer's Responses to Questions

**Comments to the Author**

1. If the authors have adequately addressed your comments raised in a previous round of review and you feel that this manuscript is now acceptable for publication, you may indicate that here to bypass the “Comments to the Author” section, enter your conflict of interest statement in the “Confidential to Editor” section, and submit your "Accept" recommendation.

Reviewer #1: (No Response)

2. Is the manuscript technically sound, and do the data support the conclusions?

Reviewer #1: Yes

3. Has the statistical analysis been performed appropriately and rigorously? 

Reviewer #1: Yes

4. Have the authors made all data underlying the findings in their manuscript fully available?

Reviewer #1: Yes

5. Is the manuscript presented in an intelligible fashion and written in standard English?

Reviewer #1: Yes

6. Review Comments to the Author

Reviewer #1: This study highlights the merits of video triage.

1. Regarding the survey question for call handlers, "Are you satisfied with the call in general". The Odds ratio of 1.95 is correctly derived but it is misleading. The odds of answering "to a very high extent" for the video triage is only 153/508 or 0.3, while the odds for the telephone triage is 78/507 or 0.15. The individual odds are small but when divided gives an odds ratio of 1.95. I suggest adding another odds ratio but this time comparing the odds of of answering both "to a very high extent" and "to a high extent". The odds will be higher for each group (about 6.3 and 3.0) and the odds ratio will be about 2.1.

2. Kindly check the range of ages of the 7 call handlers interviewed. It is written that the age ranged from 11-37. 11 is most probably erroneous.

7. PLOS authors have the option to publish the peer review history of their article (what does this mean?). If published, this will include your full peer review and any attached files.

Reviewer #1: No

---

## [Editor Report · Acceptance letter]

5 Apr 2022

PONE-D-21-08760R1 

“We can’t do without it”: parent and call-handler experiences of video triage of children at a medical helpline 

Dear Dr. Gren:

I'm pleased to inform you that your manuscript has been deemed suitable for publication in PLOS ONE. Congratulations! Your manuscript is now with our production department. 

Kind regards, 

on behalf of

Dr. Dylan A Mordaunt 

Academic Editor

PLOS ONE